# Obstructed Defecation Syndrome: Analysis of the Efficacy and Mid-Term Quality of Life of an Innovative Robotic Approach

**DOI:** 10.3390/healthcare12191978

**Published:** 2024-10-04

**Authors:** Mauro Cervigni, Andrea Fuschi, Andrea Morciano, Lorenzo Campanella, Antonio Carbone, Michele Carlo Schiavi

**Affiliations:** 1Female Pelvic Medicine & Robotic Reconstructive Surgery Center, Department of Urology, Università “La Sapienza”, ICOT Polo Pontino, 00161 Rome, Italy; mauro.cervigni@libero.it (M.C.); andreafuschi@gmail.com (A.F.); antonio.carbone@uniroma1.it (A.C.); 2Department of Obstetrics and Gynaecology, “Pia Fondazione Cardinale G. Panico”, 73039 Tricase, Italy; drmorciano@gmail.com; 3Urogynecologic Unit, Pertini Hospital, 00157 Rome, Italy; lorenzo.campanella@aslroma2.it

**Keywords:** robotic surgery, obstructed defecation syndrome, rectal prolapse, rectal wall plication

## Abstract

**Background:** The goal of our research is to demonstrate how the combination of Rectal wall Plication (RP) and robotic Ventral Mesh Rectopexy (VMR) results in a safe and effective operation that provides superior outcomes for patients with Obstructed Defecation Syndrome (ODS). **Methods:** In a total of 78 women with ODS with posterior compartment prolapse, 30 had VMR whereas 33 received VMR plus RP. We assessed VMR and VMR + RP’s efficacy and safety, as well as their influence on quality of life and sexual function. **Results:** At the median follow-up, both groups’ POP-Q categorization scores for the posterior compartment decreased (*p* < 0.001). In terms of quality of life, the PISQ-12 showed an increase in sexual quality (30.12 ± 7.12 vs. 35.98 ± 5.98 in the VMR group and 29.65 ± 6.45 vs. 29.65 ± 6.45 in the VMR + RP group, *p* = 0.041). In the VMR + RP group, the number of sexually active patients with at least two sexual interactions per month rose (*p* = 0.033). At the median follow-up, the ODS score values differed significantly (7.11 ± 1.65 vs. 1.88 ± 1.89, *p* = 0.013). **Conclusions:** The combination of rectal wall plication and ventral mesh rectopexy may result in improved bowel function and quality of life.

## 1. Introduction

Rectocele, defined as a protrusion in the posterior vaginal wall caused by an outpouching of the anterior wall of the rectum through a compromised rectovaginal fascia, is often a component of a tricompartmental defect in pelvic organ prolapse (POP) [1,2]. The real incidence of rectocele is not known, though asymptomatic posterior compartment prolapse has been documented in roughly 40% of parous women [3]. When symptomatic, posterior compartment prolapse manifests with obstructed defecation syndrome (ODS), a clinical condition characterized by difficulty in feces evacuation despite the absence of mechanical impediment, often associated with symptoms such as tenesmus, the need to digitate vaginally or anally, post-defecatory soiling, perineal pain, and, in rare cases, fecal incontinence [4,5,6].

The choice of treatment depends on the severity of the syndrome, the symptoms experienced by the patient, and the impact on the patient’s quality of life. Although symptoms can be treated with a conservative first approach, this is not always the best therapeutic choice, and in such cases, surgery is still a viable option to explore [7].

Moving toward the evolution of the surgical technique to treat ODS, an approach being widely spread among European colorectal surgeons is the transanal approach. Despite being regarded as a surgery with a high rate of success (72%), the postoperative complications remain significant [8,9,10,11]. Gynecologists’ preferred treatment method is posterior colporraphy, which commonly includes levator muscle plication. Although the transvaginal method is a safe surgery with a low complication rate, functional outcomes are extremely contradictory [12].

As explained in the European guidelines [13], the anal sphincter’s functioning status determines whether surgical or non-surgical treatment is recommended. All of the panelists oppose surgery in the context of pelvic floor dyssynergia. The transanal technique is thought to be problematic if anal sphincter function is inadequate because of the possibility of additional worsening in anal continence, and ventral rectopexy (VRP) is preferable [14].

Nowadays, Robotic ventral rectopexy, with fresh and upgraded ergonomics and accuracy, includes the mobilization of the rectum down to the level of the levator muscles and its fixation to the sacral promontory by using sutures or staples. Furthermore, the insertion of a mesh while performing rectopexy is a common practice, its placement anteriorly or around the rectum makes it a very reliable technique [15,16,17,18,19,20]

According to the recent literature, which supported the emerging role of Robotic Ventral Rectopexy and also our previous study in which we employed vaginal plication during sacrocolpopexy in patients with severe posterior vaginal prolapse [21], we further assessed the role of plicating the rectal wall to restore the anatomy and function of the rectum in patients suffering from posterior compartment prolapse and ODS, and in contrast to the transanal technique, no excision was performed.

The issue we posed ourselves was whether the combination of Rectal wall Plication (RP) and robotic Ventral Mesh Rectopexy (VMR) would result in a safe and successful procedure that might deliver better results for patients with Obstructed Defecation Syndrome (ODS). In conclusion, thanks to the notable results we obtained in terms of quality of life and reduction in symptomatology, we were able to reach the goal of our study, which is to prove how the simultaneous plication of the rectal wall in combination with robotic mesh rectopexy would result in a safe and effective treatment for the patients suffering from rectal prolapse and obstructed defecation syndrome and would provide better outcomes in terms of bowel function and quality of life.

## 2. Materials and Methods

From January 2018 to December 2021, 78 women affected by ODS and posterior compartment prolapse were referred to the Dept. of Surgery/Urology at “La Sapienza” Univ. ICOT Polo Pontino Hospital and Dept. of Urogynecology and Pelvic Floor Reconstructive Surgery of the Sandro Pertini Hospital of Rome and they were enrolled for the study.

All data were retrospectively evaluated from a collected urogynecological internal database. Therefore, this study is a retrospective study comparing the analysis of the two different techniques’ results. The Institutional Review Board (IRB) approved the study. Informed written consent was obtained from all women. The research was conducted according to Good Clinical Practice Guidelines.

The inclusion criteria were as follows: patients aged between 18 and 75 years, presence of ODS and rectocele ≥ II stage, according to Pelvic Organ Prolapse Quantification System, POP-Q classification, without coexisting significant anterior or apical compartment prolapse and without clinical or latent stress urinary incontinence (SUI).

Women with concomitant surgery of uterine prolapse, cystocele, or urinary incontinence were excluded. The exclusion criteria were also malignancies, degenerative neurological diseases, previous pelvic radiotherapy, pregnancy state, megacolon, bowel inflammatory disease, pelvic floor dyssynergia or anal sphincter deficiency, and contraindications to surgery or anesthesia. In the presence of sphincter contractile deficiency or dyssynergia, the patients were first treated with pelvic floor rehabilitation and then, after a re-evaluation with anorectal manometry, scheduled for surgery.

The procedures were ever performed by the same surgeon (MC) who had previously performed more than 100 interventions completing the learning curve.

The minimum follow-up for these patients was 18 months.

Preoperative assessment was conducted using a standardized questionnaire and a clinical examination. The questionnaire consisted of a history of bowel movements, questions about obstructed defecation, need for vaginal/perineal digitation and prolapse protrusion symptoms (foreign body feeling in the introitus), and a history of anal incontinence and sexual function.

All patients were candidates for surgical intervention after the failure of medical and dietary therapy and after a complete radiological and functional study.

Magnetic resonance (MR) or X-rays (RX) defecography and anorectal manometry were always performed to assess the presence of rectocele and rectal intussusceptions in patients who were unable to empty. These examinations were also helpful to identify the presence of pelvic floor dyssynergia or anal sphincter deficiency, which are criteria for exclusion from the surgical treatment.

The clinical examination consisted of a proctological and gynecological examination in the supine lithotomy or lateral position during the maximal Valsalva maneuver. POP-Q measurement according to the guidelines of the International Continence Society (ICS) was performed.

The degree of posterior compartment defects was evaluated by the POP-Q System under maximum straining effort, with the patient in the lithotomy position.

The symptoms of ODS that are perceived by the patients were evaluated before and after the intervention using the ODS (Obstructed Defecation Syndrome) score, which analyzes constipation using 5 items.

The prolapse quality of life questionnaire (P-QoL) was used to quantify the impact of prolapse symptoms on QoL. The Pelvic Organ Prolapse/Urinary Incontinence Sexual Questionnaire short form (PISQ-12), Pelvic Floor Disability Index (PFDI-20), and Pelvic Floor Impact Questionnaire (PFIQ-7) were administered to evaluate quality of life and sexual function before surgical intervention and at median follow up [22,23,24,25].

Before surgery, all patients were given low molecular–weight heparin to prevent venous thromboembolism. Short-term antibiotic prophylaxis was performed 30 min before surgery.


**Surgical technique**


Robotic Ventral Mesh Rectopexy is a procedure utilizing da Vinci^®^ technology, involving the mobilization of the front of the rectum all the way to the pelvic floor, and fixation of the front of the rectum and pelvic floor to the bony sacral promontory with a mersilene mesh. This straightens the rectum and restores its tubular shape.

In this technique, the nerves to the rectum that enter from the back and side are preserved with mobilization only from the front and right side of the rectum, with careful preparation of the rectal fascia and muscular layer. This step is of utmost importance for the preservation of nerve endings, leaving the back of the rectum for the normal maintenance of rectal functionality intact and reducing the risk of hindgut neuropathy and resultant constipation. The mersilene mesh is thought to result in a lower recurrence rate than sutured rectopexy whilst avoiding long-term complications

The surgical technique of rectal wall plication is an innovative procedure consisting of the placement of three sutures utilizing monofilament slow resorption resorbable threads (PDS II 2-0) for the plication in the midline of the anterior wall of the rectum starting 1 cm above the perineal body and subsequently at the distance of 1.2–1.5 cm before fixing the mesh for the rectopexy. This allows a decrease in the caliber of the rectum with greater stabilization.


**Operative and postoperative assessment**


After surgery, a bladder catheter and a vaginal pack were positioned and were removed after 48 h. Operative time, blood transfusions, spontaneous voiding, perioperative complications, postoperative hospital stay, and postoperative complications (early within 30 days and late after 30 days) were considered.

The following questionnaires were administered after surgery at median follow-up: P-QoL, PFDI-20 PFIQ-7, and PISQ-12. A clinical examination and the compilation of questionnaires were performed at least 18 months after surgery.

Administering these questionnaires is standard care in our Urogynecology and Proctology clinic. The same team performed surgery and postoperative evaluation.

The objective cure for prolapse was defined as a remaining posterior defect of stage 0-I, evaluated by POP-Q classification under maximum straining effort with the patient in the lithotomy position. Recurrence of prolapse was defined as stage II or higher, based on the POP-Q classification.

Patients with a significant worsening of defecatory function or with a suspicion of a prolapse recurrence were subjected to an MR defecography. Patients who complained of fecal incontinence were studied with anorectal manometry.

The ODS (Obstructed Defecation Syndrome) score after surgery to evaluate constipation symptoms was used.

The primary endpoint was to evaluate the effectiveness and the safety of RVMR and RVMR + RP at median follow-up.

The secondary endpoint was to evaluate the impact on quality of life and sexual function of these two surgical procedures.

We analyzed the incidence of each event to define its statistical significance using Fisher’s exact test.

We analyzed the incidence of every event to define its statistical significance using Fisher’s exact test. The odds ratio (OR) and 95% confidence intervals (CIs) were calculated for every comparison. Normality tests (D’Agostino and Pearson test) were performed to determine whether data were sampled from a Gaussian distribution. The *t*-test and Mann–Whitney U test were used to compare continuous parametric and non-parametric variables (when data do not fit into the normal distribution), respectively. Correlations between numerical parameters were computed using the Spearman rank correlation coefficient. A matched *t*-test was applied to determine changes in the score of questionnaires (ODS Score, P-QoL, PFDI-20 PFIQ-7, and PISQ-12) values.

All analyses were conducted using the Statistical Package for the Social Sciences (SPSS) 22.0 for Mac (SPSS, Chicago, IL, USA). Significance was set at a *p*-value of <0.05.

## 3. Results

In total, 78 pats were evaluated. Nine women refused the treatment, three were excluded because they had poor performance status (ECOG > 2), and three were lost to follow-up; hence, 63 women were analyzed. A total of 63 patients were analyzed.

Demographic Clinic and Pathological characteristics were shown in Table 1; no significant differences were reported between the two groups. No significant difference in terms of operating time occurred (132 min vs. 123 min; *p* = 0.121).

All included patients (63/63) showed the simultaneous presence of rectocele and recto-anal or recto-rectal intussusceptions at preoperative defecography.

Similar results were shown in terms of blood loss, intraoperative complications, ureteral injuries, bowel injuries, bladder injuries, hemoperitoneum, and rectal abscesses. One patient (group A) and one patient (group B) needed a blood transfusion during the surgical intervention. There were no ureteral, bladder, or intestinal lesions in the two groups.

According to symptomatic evaluation and analysis of postoperative complications, the VMR + RP group only totalized two cases of constipation and one at the median follow-up time, whereas the VMR group had 12 cases of constipation in the first postoperative month, which decreased to 6 cases at the median follow-up time (*p* = 0.023, Table 2).

The median follow-up was 23 months (18–26).

The POP-Q classification score for the posterior compartment showed a significant average decrease for both groups (*p* < 0.001) at median follow-up and there was not a significant difference between the two groups. Five (16.6%) women in the VMR group and 0 (0%) in the VMR + RP group used vaginal digitation (*p* = 0.041) at least 18 months after surgery. Questionnaires assessing quality of life associated with gastrointestinal symptoms did not demonstrate significant differences at median follow-up (Table 3).

In the VMR + RP group, the number of sexually active patients after surgery with at least two sexual intercourses per month increased (*p* = 0.033) and consequently, the PISQ-12 showed an improvement in the quality of sexual life after at least 18 months of follow-up (30.12 ± 7.12 before surgery and 35.98 ± 5.98 after surgery in VMR group vs. 29.65 ± 6.45 before surgery and 29.65 ± 6.45 after surgery in VMR + RP group, *p* = 0.041) (Table 3).

At PGI-I evaluation, no significant differences occurred (Table 4).

At 6 months of follow-up, the ODS score showed no significant difference (6.31 ± 2.69 in the VMR group vs. 2.37 ± 1.59 in the VMR + RP group, *p* = 0.11). Contrary to that, at the follow-up time of 12 and 18 months, ODS score values were significantly different: (5.11 ± 1.88 vs. 1.23 ± 1.14 *p* = 0.03; 7.22 ± 1.54 vs. 1.57 ± 1.14 *p* = 0.02). The other values of the intermediate follow-up are shown in Table 5.

Consequently, at the median follow-up time, the ODS score values were significantly different: (7.11 ± 1.65 vs. 1.88 ± 1.89, *p* = 0.013). The other values of the intermediate follow-up (12 and 18 months) are shown in Table 5.

## 4. Discussion

To our knowledge, this is the first study reporting simultaneous rectal wall plication with the combination of robotic mesh rectopexy in patients suffering from ODS and rectocele.

This concept was inspired by our recent research [21], where we hypothesized that patients with severe posterior compartment prolapse would benefit from a laparoscopic sacral colpopexy plus plication of the vaginal fascia in order to help restore their anatomical and functional outcomes.

In fact, we hypothesized that restoring the rectal wall architecture would result in decreased stool transit time and constipation postoperatively, as well as improved bowel function and quality of life in these patients.

Laplace’s rule, as the foundation of our clinical reasoning, explains how strength and tightness of the rectal lumen would provide us with more contractile strength to ensure fecal emptying [4].

According to our findings, this new technique involving rectal plication and mesh rectopexy had fewer postoperative complications than the mesh-only group, as well as a significant improvement in quality of life, sexual activity, and ODS symptomatology, which was significantly reduced at the postoperative median follow-up.

Colorectal surgeons commonly performed transanal access to treat ODS, with different degrees of effectiveness. Arnold et al. [26] observed poor postoperative outcomes, with 54% of patients complaining of constipation. According to Roman et al. [27], functional results declined with increasing periods of follow-up, reaching a recurrence rate of 50% after 5.5 years. Furthermore, roughly one-third of female patients experienced a new beginning of anal incontinence.

Regarding the clinical outcome, a retrospective multicentric study of the Italian Society of Colo-Rectal Surgery [28] found that at 18 months after STARR, 55% of the patients still had at least three symptoms of ODS, and that 19% of the cases needed a reintervention due to either postoperative complications or recurrence of symptoms. Dealing with postoperative complications, defecation urgency is still the most frequent longer-term adverse effect, occurring in up to 10% of patients on average. Rectal stenosis is an uncommon complication that usually affects fewer than 1% of patients over the longer term (12 months or more) and less than 2% of patients usually develop longer-term discomfort.

Anastomotic dehiscence and postoperative sepsis, although they are uncommon side effects, could occur in patients undergoing rectal resection.

Patient global satisfaction ratings, while not always constant, generally indicated an acceptable outcome for around 73–80% of patients although, in around 68–76% of patients, there was an inconsistent decrease of 53–91% in the ODS score for obstructed defecation syndrome.

Due to the high postoperative complication rate in the transanal approach, robotic ventral rectopexy was proposed. Thanks to the absence of resection of the rectal mucosa to treat the primary component and the consequent neuromuscular condition underlying the development of ODS syndrome, it could become the first-line treatment for patients suffering from rectocele and associated ODS.

According to De Hoog et al. [29] and Mehmood et al.’s recent clinical experiment [30], this procedure is associated with lower constipation and fecal incontinence compared to perineal procedures [15,31,32]. Furthermore, a reported median Cleveland Clinic Constipation Score (CCCS) gain of 3.2 points after robotic surgery and Wexner incontinence score were noticeably lower than that of the other procedures, with a noteworthy improvement in blocked defecation [33].

Even in patients undergoing rectopexy, several studies [34,35] found radiological recurrence of rectocoele, with rates ranging from 0% to 15% together with the rate of postoperative constipation where the findings were 12–22.5%.

According to D’Hoore et al. [36], our treatment solely comprised anterior rectal wall mobilization. As a result, posterior and lateral rectum mobilization were no longer required in order to reduce the risk of nerve injury. According to studies [37,38], constipation following rectal resection and posterior suture rectopexy is less prevalent than after posterior rectopexy without resection.

In our study, as demonstrated by our findings, this new technique involving rectal plication and mesh rectopexy had fewer postoperative complications than the control group, emphasizing how postoperative constipation, in particular, was significantly lower in the plication group than in the mesh-only group.

At median follow-up, the POP-Q categorization score for the posterior compartment decreased significantly for both groups with no significant difference between the two groups.

After 23 months, the number of sexually active patients in the VMR + RP group increased dramatically, and the QoL surveys and ODS score values showed incredible results.

It is clear that, if these findings are validated by additional data, the results we obtained may be useful to doctors in the future as new clinical practice recommendations, as we provided fresh evidence on the issue of prolapse-related ODS, with the goal of delivering new insights into surgical techniques capable of treating posterior compartment prolapse and ODS-related symptomatology.

However, randomized studies and longer follow-up trials are required to better understand the potential function of this new surgical method and, ultimately, to validate the safety and improvement in quality of life seen in the current research. In the future, further technological innovations and devices might be added in order to make this technique more efficient, always paving toward mininvasivity, safety, and efficacy.

Our study is not free of limitations. First, because this is retrospective research, there is the possibility of selection bias and the absence of randomization. Second, in this trial, we were unable to evaluate long-term outcomes, such as recurrence rates. Finally, the sample size is small. It is possible that in a larger population, we could have seen differences in anatomical outcomes as well.

## 5. Conclusions

Finally, at the follow-up time of 18 months postoperatively, the ODS score was significantly lower from the beginning value in both groups who received mesh rectopexy alone and mesh rectopexy together with rectal plication, possibly as a result of a long-term readapatation of muscle fibers.

Further randomized studies are necessary to confirm the data on the efficacy and safety of the technique combining robotic rectopexy with mesh with simultaneous plication of the rectal wall.

## Figures and Tables

**Table 1 healthcare-12-01978-t001:** Clinic Pathological characteristics and Surgical procedures in 63 patients.

*Clinical Variables*	VMR (30)	VMR + RP (33)	*p*
**Mean Age (SD)**	62.34 (4.75)	60.89 (4.09)	0.21
**Median Vaginal Delivery (range)**	2 (1–5)	2 (1–4)	0.67
**Mean BMI (SD)**	27.34 (3.82)	27.89 (4.03)	0.21
**Menopause Status (%)**	25 (83%)	30 (90%)	0.55
**Smokers (%)**	4 (13%)	6 (18%)	0.45
** *Pelvic Organ Prolapse Stage (posterior compartment)* **			
**Stage II (%)**	12 (40%)	12 (36%)	0.63
**Stage III (%)**	12 (40%)	15 (45%)	0.23
**Stage IV (%)**	6 (20%)	6 (18%)	0.78
** *Previous Surgical Procedure* **			
**Hysterectomy (%)**	1(3%)	1 (3%)	0.34
**Bilateral Adnexectomy (%)**	2 (6%)	1 (3%)	0.22
**Shull Suspension (%)**	1 (3%)	1 (3%)	0.71
**Abdominal Sacrocolpopexy (%)**	1 (3%)	1 (3%)	0.59
**Anterior Colphorraphy (%)**	1 (3%)	1 (3%)	0.88
**Posterior Colphorraphy (%)**	2 (6%)	3 (9%)	0.29
**Continence Surgery (%)**	2 (6%)	1 (3%)	0.65

**Abbreviations:** SD: Standard Deviation; BMI: Body Mass Index; TOT: transobturator tape.

**Table 2 healthcare-12-01978-t002:** Complications in 63 patients after surgery (median follow-up) *.

Complications	VMR (30) 1 Month	VMR (30) Median FU	VMR + RP (33) 1 Month	VMR + RP (33) Median FU	*p*
**Rectal stenosis (%)**	0 (0)	0 (0)	0 (0)	0 (0)	ns
**Proctalgia, pain (%)**	1 (3.3)	0 (0)	2 (6)	1 (3)	ns
**Obstructed defecation syndrome after surgery (%)**	1 (3.3)	1 (3.3)	1 (3)	1 (3)	ns
**Tenesmus (%)**	0 (0)	0 (0)	1 (3)	0 (0)	ns
**Post-defecatory soiling (%)**	1 (3.3)	0 (0)	1 (0)	0 (0)	ns
**Dyspareunia (%)**	1 (3.3)	0 (0)	0 (0)	0 (0)	ns
**Fecal Urgency (%)**	4 (13.3)	1(3.3)	2 (6)	1 (3)	ns
**Constipation (%)**	12 (40)	6 (20)	2 (6)	1 (3)	0.023
**Fecal Incontinence (%)**	0 (0)	0 (0)	0 (0)	0 (0)	ns
**Recto-Vaginal Fistula (%)**	0 (0)	0 (0)	0 (0)	0 (0)	ns
**Difficult voiding (%)**	0 (0)	0 (0)	0 (0)	0 (0)	ns
**Overactive Bladder (%)**	0 (0)	0 (0)	1 (3)	0 (0)	ns
**Stress urinary incontinence (%)**	0 (0)	0 (0)	0 (0)	0 (0)	ns
**Urge urinary incontinence (%)**	0 (0)	0 (0)	0 (0)	0 (0)	ns
**Recurrent Urinary Tract Infections (%)**	1 (3.3)	1 (3.3)	1 (3)	1 (3)	ns

*: 23 months (18–26).

**Table 3 healthcare-12-01978-t003:** Pre and Postoperative (Bp) POP-Q score Classification, Quality of Life and Sexual Function.

Variables	Preoperative VMR (30)	Median Follow-Up	*p*	Preoperative VMR + RP (33)	Median Follow-Up	*p*	VMR vs. VMR +RP
**Posterior Compartment (Bp)**	1.52 ± 1.85	2.63 ± 0.34	<0.001	1.48 ± 0.42	−2.41 ± 0.64	<0.001	0.45
**Vaginal Digitation (%)**	21 (70)	5 (16.6)	0.034	23 (69.7)	0 (0)	<0.001	0.041
**Vaginal Bulge (%)**	25 (83.3)	1 (3.3)	<0.001	25 (75.7)	1 (3)	<0.001	0.76
**P-QoL**	65.85 ± 17.12	32.03 ± 8.57	0.004	64.22 ± 16.23	27.54 ± 8.56	< 0.001	0.54
**PFDI-20**	146.34 ± 65.27	41.76 ± 27.89	<0.001	144.87 ± 63.88	38.87 ± 29.65	<0.001	0.67
**PFIQ-7**	71.34 ± 54.87	12.76 ± 17.98	<0.001	70.98 ± 53.79	11.74 ± 23.65	<0.001	0.50
**Sexual Activity (%) ***	12 (40)	19 (63.3)	0.047	11 (33.3)	25 (75.7)	0.007	0.033
**PISQ-12**	30.12 ± 7.12	35.98 ± 5.98	0.034	29.65 ± 6.45	29.65 ± 6.45	< 0.001	0.041

**Abbreviations:** POP-Q score: Pelvic Organ Prolapse Quantification score; P-QoL: prolapse quality of life questionnaire; PFDI-20: Pelvic Floor Disability Index; PFIQ-7: Pelvic Floor Impact Questionnaire; PISQ-12: Pelvic Organ Prolapse/Urinary Incontinence Sexual Questionnaire short form. *: Sexual activity was not advised until one month after surgery; at least two sexual intercourses a month.

**Table 4 healthcare-12-01978-t004:** Patient impression of global improvement (PGI-I) at the Median Follow Up.

Variables	VMR (30)	VMR +RP (33)	*p*
**1: very much better**	21 (70%)	24 (72%)	NS
**2: much better**	3 (10%)	5 (15%)	NS
**3: a little better**	3 (10%)	3 (9%)	NS
**4: no improvement**	3 (10%)	1 (3%)	NS
**5: a little worse**	0	0	NS
**6: much worse**	0	0	NS
**7: very much worse**	0	0	NS
**Success**	24 (80%)	29 (87%)	NS

**Table 5 healthcare-12-01978-t005:** ODS Score.

Mean ODS Score	VMR 30	VMR + RP 33	*p*
**Preoperative **	23.17 ± 4.82	22.23 ± 3.87	0.34
**Postoperative **			
**6 mos **	6.31 ± 2.69	2.37 ± 1.59	0.11
**12 mos **	5.11 ± 1.88	1.23 ± 1.14	0.03
**18 mos **	7.22 ± 1.54	1.57 ± 1.14	0.02
**Median follow up **	7.11 ± 1.65	1.88 ± 1.89	0.013

## Data Availability

Data is contained within the article.

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
