# Peer review of "Obstructed Defecation Syndrome: Analysis of the Efficacy and Mid-Term Quality of Life of an Innovative Robotic Approach"

_healthcare, 2024, doi:10.3390/healthcare12191978_

Round 1
Reviewer 1 Report
Comments and Suggestions for Authors
Manuscript Title: "Obstructed Defecation Syndrome: analysis of the efficacy and mid-term quality of life of an innovative robotic approach" by Mauro Cervigni, Andrea Fuschi, Andrea Morciano, Lorenzo Campanella, Antonio Carbone, and Michele Carlo Schiavi
This manuscript presents a well-conducted study analyzing the efficacy and mid-term quality of life outcomes of a novel robotic surgical approach for Obstructed Defecation Syndrome. The authors have meticulously detailed their methodology and have provided substantial evidence supporting the benefits of combining Rectal wall Plication with robotic Ventral Mesh Rectopexy. The results indicate a significant improvement in both bowel function and quality of life, making this a valuable contribution to the field. The study is comprehensive, and the writing is clear, making it accessible to both clinicians and researchers.
Specific Comments:
- The introduction is overly lengthy and could benefit from being more concise. It is advisable to shorten it into several distinct paragraphs, ensuring that each paragraph serves a clear purpose. In the penultimate paragraph, please explicitly outline what is known so far and what remains unknown in the field, and identify the specific questions that this study aims to address. In the final paragraph, briefly mention some of the key findings of the study to set the stage for the detailed discussion that follows.
- Where statistical significance is present, it would be beneficial to visually represent these results using bar diagrams. This will help to enhance the clarity of the data and make the statistical differences more evident to the reader.
- The limitations of the study are not sufficiently detailed. It is crucial to elaborate on the potential limitations of your research, including any aspects related to the study design, patient selection, and the generalizability of the findings.
- A dedicated section in the discussion that focuses on clinical practice recommendations would be extremely valuable. This section should interpret the findings in the context of their practical application in clinical settings, providing guidance to clinicians on how the study results could influence their practice.
- Additionally, a separate discussion on the knowledge gaps and recommendations for future research is essential. This will not only highlight the study's contributions but also pave the way for future studies to build on these findings.
- The references are not uniformly formatted, which detracts from the overall professionalism of the manuscript. Furthermore, more than 50% of the references are older than five years. It is recommended to update the reference list with more recent literature to reflect the current state of research in the field.
- The methodology section could benefit from more detail regarding the statistical methods used.
Author Response
Dear Editor,
Thank you for your revisions. Here is the point-by-point response.
1. We slightly adjusted the opening part in response to your ideas. We will not separate the introduction in paragraphs since we believe it will result in a more confusing and less clear section of the text.
2. In our opinion, we use graphical tables because we believe they help the reader to understand the study and its findings.
3. We changed the part on the study's limitations.
4. As previously said, we disagree with the splitting of the sections into paragraphs, but we took your advice and attempted to implement the discussion.
5. As previously said, we followed you suggestions and attempted to enhance the discussion.
6. We formatted the references section and incorporated more recent citations.
7. We personally did the statistical study and employed the approach we stated, which we believe is sufficient to provide a thorough understanding of the statistical process below.
Thanking you in advance for your cooperation, we wish you a nice day.
Kind regards,
Michele Carlo Schiavi
Reviewer 2 Report
Comments and Suggestions for Authors
I appreciate the opportunity to review this interesting article. It is a retrospective research that seeks to evaluate the effectiveness of robotic surgery approach as a treatment for Obstructed Defecation Syndrome. The introduction contains important information to present the rationale of the research, as well as an appropriate theoretical framework regarding the SDG, the methodology is clearly explained, and the results presented are discussed in detail. I have only a few observations:
-Bibliographic references are missing to support lines 61-65.
-It is important to avoid speculative or evaluative statements in a scientific research article, e.g. line 61 mentions "with better ergonomics and precision", or lines 300-301 talk about "good number of patients" or "strong conclusions".
-Several abbreviations are used without mentioning their meaning (it is important to mention the meaning the first time it appears in the text), for example, MR, RX, ICS, RVMR, RP.
-It is necessary to place references to the scales/indices used in the study (in the methodology section, lines 122-126).
-In the limitations, it is necessary to modify the text regarding the number of patients. The sample size is not defined as good or bad, it is defined as sufficient or insufficient, referring to the sample size calculation based on the objectives of the research. In studies where the sample size is not calculated, it is important to highlight this weakness, and one can also highlight the fact that it is the study with the largest number of patients analyzed with respect to the topic being discussed. The conclusions are not "weak" or "strong", they are only conclusions that should be limited or restricted to the results obtained.
-I think it is important that the authors add a conclusion section.
Author Response
Dear Editor,
Thank you for your revisions. Here is the point-by-point response.
1. We inserted the bibliographic reference.
2. We corrected the errors.
3. We fixed the error and included explanations for the abbreviations.
4. We inserted the bibliographic reference.
5. We adjusted the statement to make it more appropriate.
6. We included a concluding section.
Thanking you in advance for your cooperation, we wish you a nice day.
Kind regards,
Michele Carlo Schiavi
Reviewer 3 Report
Comments and Suggestions for Authors
Dear authors
Congratulations for your very good clinical work and scientific results.
Minor corrections are needed to improve the quality of the work.
I suggest that you attach an illustration of this new robotic operating technique, if you are able.
line 194: One patient in group A and 1 patients in group B needed a blood transfusion…. Please, define group A and group B in the brackets
Please correct word 1 patients- 1 patient
line 226… Tor our knowledge, correct to our
line 253.. withe, correct to with
Since this is a new technology, it would be great if you could make a graphic abstract.
Best Regards
Author Response
Dear Editor,
Thank you for your revisions. Here is the point-by-point response.
1. I don't think it's feasible to provide an illustration of the process, but we will attempt.
2. We rectified the grammatical mistakes and incorporated your suggestions.
3. We rectified the error.
4. We rectified the error.
5. As we previously stated for the illustration of the approach, we will examine if it is feasible to build a graphical abstract, although we are unsure about it.
Thanking you in advance for your cooperation, we wish you a nice day.
Kind regards,
Michele Carlo Schiavi
Round 2
Reviewer 1 Report
Comments and Suggestions for Authors
Minimal changes have been made to the text of the manuscript, and the majority of the critiques have not been adequately addressed. Therefore, I recommend that this manuscript in its current form should not be accepted for publication.
Author Response
Dear Editor,
Thank you for your revisions. Here is the point-by-point response.
We apologize for not fully responding to your previous comments, but if you compare the most recent and previous version of the manuscript, you will notice that we made various revisions and modifications based on your ideas. In this iteration, we endeavored to follow your suggestions more closely and create a version that better suited your requests.
Thanking you in advance for your cooperation, we wish you a nice day.
Kind regards,
Michele Carlo Schiavi